# Body Fat Percentage, Body Mass Index, Fat Mass Index and the Ageing Bone: Their Singular and Combined Roles Linked to Physical Activity and Diet

**DOI:** 10.3390/nu11010195

**Published:** 2019-01-18

**Authors:** David J. Tomlinson, Robert M. Erskine, Christopher I. Morse, Gladys L. Onambélé

**Affiliations:** 1Musculoskeletal Sciences and Sport Medicine Research Centre, Manchester Metropolitan University, Crewe CW1 5DU, UK; c.morse@mmu.ac.uk (C.I.M.); g.pearson@mmu.ac.uk (G.L.O.); 2Research Institute for Sport and Exercise Sciences, Liverpool John Moores University, Liverpool L3 3AF, UK; r.m.erskine@ljmu.ac.uk; 3Institute of Sport, Exercise and Health, University College London, London W1T 7HA, UK

**Keywords:** nutrition, aging, adiposity, physical activity, bone, inflammation

## Abstract

This study took a multi-analytical approach including group differences, correlations and unit-weighed directional z-score comparisons to identify the key mediators of bone health. A total of 190 participants (18–80 years) were categorized by body fat%, body mass index (BMI) and fat mass index (FMI) to examine the effect of differing obesity criteria on bone characteristics. A subset of 50 healthy-eating middle-to-older aged adults (44–80 years) was randomly selected to examine any added impact of lifestyle and inflammatory profiles. Diet was assessed using a 3-day food diary, bone mineral density (BMD) and content (BMC) by dual energy x-ray absorptiometry in the lumbar, thoracic, (upper and lower) appendicular and pelvic areas. Physical activity was assessed using the Baecke questionnaire, and endocrine profiling was assessed using multiplex luminometry. Obesity, classed via BMI, positively affected 20 out of 22 BMC- and BMD-related outcome measures, whereas FMI was associated with 14 outcome measures and adiposity only modulated nine out of 22 BMC- and BMD-related outcome measures. Whilst bivariate correlations only linked vitamin A and relative protein intake with BMD, the Z-score composite summary presented a significantly different overall dietary quality between healthy and osteopenic individuals. In addition, bivariate correlations from the subset revealed daily energy intake, sport-based physical activity and BMI positive mediators of seven out of 10 BMD sites with age and body fat% shown to be negative mediators of bone characteristics. In conclusion, whilst BMI is a good indicator of bone characteristics, high body fat% should also be the focus of osteoporosis risk with ageing. Interestingly, high BMI in conjunction with moderate to vigorous activity supplemented with an optimal diet (quality and quantity) are identified as positive modulators of bone heath.

## 1. Introduction

Bone loss in men and women is a consequential process of ageing [1] with mean variations estimated to range between 0.86% and 1.12% bone loss per year in elderly men and women [2]. However, at its extreme, age-related bone loss can lead to osteoporosis, a condition characterized by an increased risk of bone fractures [3] through a reduction in bone tissue altering the structural integrity/architecture [4] and even leading to premature mortality [5]. Previous research has identified independent accelerants of poor bone health such as decreased physical activity (PA) caused by the reduction of mechanical loading/stress placed on bone [6], poor quality and inadequate nutritional intake [4] and obesity [7]. Whilst existing research has independently examined how each of these lifestyle behaviors influence bone health, questions remain on the cumulative effect of dietary content/quality, type of PA in conjunction with age. Moreover, whether obesity definition and/or classification have any effect on the conclusion regarding bone health modulation has yet to be categorically understood, especially in a middle-to-older age adult population.

Diet and PA are two modifiable behaviors that have the potential to affect numerous systems that regulate bone homeostasis through influencing the key endocrine regulators of bone metabolism [4]. The key nutrients positively associated with bone health include calcium [8], magnesium [9], phosphorus [10], potassium [11], vitamin D (VitD) [12], Vitamin K [13], protein [14] and omega 3 fatty acids [15]. In fact, a number of dietary elements negatively influence bone health, including saturated fat [16] and Vitamin A [17]. However, the consensus within the literature focuses on two nutrients with regards to bone health, that being calcium and VitD. Calcium is the key nutrient involved in bone homeostasis due to its role in bone growth and development [4,8], with current UK guidelines recommending >700 mg/day [18] and various research studies utilizing doses as high as 1600 mg [19]. Interestingly, whilst calcium supplementation may influence bone health, it cannot be used as a replacement for prescribed estrogen, bisphosphonates, or calcitonin therapy, but only as a preventative measure when individuals are still within a normal T-score range [19]. Nevertheless, it remains to be seen if any specific interaction exists between dietary calcium and other nutrients positively associated with bone dimensional characteristics or that may aid in the absorption of calcium, such as oligosaccharides [20] and VitD [21]. Then, and only then, might the optimum effect of calcium supplementation on bone health be conclusive.

The second key nutrient is VitD, as it is purported to being crucial not only for bone health but previous research has also reported its positive association with muscle strength prominently through improved neuromuscular function [22] and stimulation of protein synthesis [23]. The current literature, however, suggests that the benefits of VitD supplementation may only be beneficial in individuals who are VitD deficient [24], especially for musculoskeletal parameters [25,26]. VitD deficiency impacts bone in two different ways, the first resulting in inadequate mineralization of the skeleton potentially causing osteomalacia, yet this may be related to primary hyperparathyroidism created by the VitD deficiency [27] and the second through negatively affecting the intestinal absorption of calcium [27]. Therefore, if conforming to the recommended daily VitD 10 µg intake [18], questions remain whether (a) a linear relationship between bone health and VitD exists where the individual is not VitD deficient [24], or (b) VitD benefits are only observed when combined with sufficient nutrients positively related to bone health (e.g., calcium, phosphorus, magnesium and Vitamin K).

As mentioned above, negative dietary contributors of bone health include high saturated fat intake [16] and Vitamin A [17]. The evidence demonstrates an inverse relationship between dietary saturated fat intake and bone mineral density (BMD) potentially due to inhibiting calcium absorption and down-regulating osteoblast formation [16]. Similarly, a high level of Vitamin A triggers the production of osteoclasts, subsequently causing bone breakdown [17]. Therefore, this demonstrates that independent associations and the interaction between nutrients need further scrutiny to aid understanding of how nutrients interact to influence bone health and, ultimately, help formulate individualized habitual nutritional guidelines.

In conjunction with habitual diet, placing mechanical load/stress on bones is known to stimulate an increase in bone formation [28] and resultant bone strength [29]. This has been shown from adolescents to the elderly [30,31]. Thus impactful PA maintains its status as an effective mechanism in combating age-related decrease in BMD. Interestingly, PA is generally grouped as one behavior in large cross-sectional or longitudinal studies in regression models. Arguably, for all and especially a middle aged or an elderly population group, PA ought to be broken down into different strands (e.g., work, leisure and sport), modalities (e.g., aerobic vs. resistance) or intensity (e.g., bowls vs. gym sessions) in order to distinguish appropriate, effective and palatable lifestyle PA interventions. However, the focus of the existing body of research on PA and BMD is between structured resistance/weight bearing and aerobic exercise [32,33]. However, the selection of a preference for modality is intuitive, as both forms of exercise elicit similar increases in spine BMD following 12–24 months of structured PA (resistance 0.8–6.8% increase [32,33,34] vs. aerobic 1.4–7.8% increase [32,35,36,37]). Interestingly, structured PA only constitutes ~3% of an individual’s waking hours if just achieving the recommended daily 30 min of moderate PA [38] (assuming 16 h awake), thus potentially missing quantifiable daily activity markers that may influence bone health. Therefore, accurate representations of an individual’s activity profile may aid in the development of detailed prediction models and sustainable prescription guidelines to prevent the escalation of bone health towards an osteoporotic profile.

Thus, the present study was spilt into two sections, with the primary aim to examine how obesity, defined through three different methods, affects bone as we age. The second aim was to take a multi-analytical approach to examining the lifestyle factors of bone mass homeostasis, ranging from habitual nutritional intake to PA. In this way, the study aimed to prioritize key identifiable areas that may aid in the reduction of ageing-associated osteoporosis risk. It was hypothesized that: (1) High adiposity would increase osteoporosis risk with age; (2) optimal dietary composition (low saturated fats, high calcium, Vitamin D, Vitamin C, oligosaccharide, protein, omega 3 and 6 fatty acids, Vitamin K, zinc, magnesium and phosphorus) would promote bone health; (3) the negative impact of high adiposity would be greater on under-loaded bone sites; (4) high levels of structured PA (more so than work- or leisure-based PA) would improve BMD; (5) endocrine profiling would be linearly associated with diet and hence bone health.

## 2. Materials and Methods

### 2.1. Participants

One hundred and ninety participants (males = 65 and females = 125) aged 18–80 years were recruited and screened prior to undertaking any assessments through a general health questionnaire, where their PA level was ascertained. The participants were split into two groups, either trained (*n* = 27) or untrained (*n* = 163), with the untrained individuals being the main focus of analyses with regard to the impact of obesity on bone health. The classification of ‘trained’ was denoted by the participant undertaking structured exercise of over 3 h per week. Primarily, the participants were categorized by three different methods of classifying obesity to determine the effect of obesity classification on bone characteristics. These were: body fat% - (Male = normal adipose (NA) <28%: high adiposity (HA) ≥28%; NA; female = NA <40%: HA ≥40%;), BMI (underweight (BMI <19), normal weight (NW; BMI ≥19–<25), overweight (BMI ≥25–<30) and obese (BMI ≥30)), and fat mass index (FMI; fat deficit male < 3, female < 5; normal male 3–6, female 5–9; excess fat male >6–9, female >9–13; obese male >9, female >13).

Secondly, to determine the effect of obesity, PA and nutrition on bone health with ageing, 50 untrained participants (males = 15 and females = 35) aged 43–80 years (see Table A1) were randomly selected to cover the body composition and age spectra, and then categorized by their bone health (normal range T-score ≥ −1.0 *n* = 42 and osteopenia T-score < −1.0 *n* = 8) for Z-score comparisons. Participants were excluded if they had changed their diet and/or PA levels in the past 12 months and if they were taking any medication related to osteoporosis/bone health. On completion of the health and PA questionnaire, their dominant arm and leg were ascertained through verbal questioning. Prior to the commencement of the study, participants gave their written informed consent and all the procedures in this study were in accordance with the Declaration of Helsinki and had approval from the Manchester Metropolitan University Ethics Committee (Ethics Committee Reference Number: 09.03.11 (ii)).

### 2.2. Measurement of Body Composition

The bone mineral content (BMC), BMD and overall body composition (both fat and lean mass) were established using a dual energy x-ray absorptiometry scanner (Hologic Discovery: Vertec Scientific Ltd., Reading, UK: see Figure 1) to accurately quantify bone characteristics and define obesity following a 12-h fasted period. Prior to the arrival of each participant, a control phantom was scanned to ensure the reliability and reproducibility of the BMC, BMD and area scan results (accepted coefficient variation of <0.6%). On arrival, the participants were given a hospital gown and asked to remove all clothing and jewelry to ensure the process was standardized between the participants. The participants were then asked to lie in the center of the scanning bed in a supine position with their head positioned in the center just inside of the scanner’s viewing field. The investigator ensured the participant’s whole body was positioned correctly to guarantee there was no contact between their trunk and appendicular mass, with their legs internally rotated (10–25°) to expose the fibula and the neck of the femur and then strapped in position using micropore tape (3M, Bracknell, Berkshire, UK) to avoid any discomfort and movement during the 7-min scanning procedure (whole body, EF 8.4 lSv). The scan results were calculated using the Hologic APEX software (version 3.3: Hologic Inc., Bedford, MA, USA) and presented in terms of the whole body lean mass, fat mass, BMC and BMD and were manually digitized using anatomical markers classifying defined body segments by their dominant and non-dominant side (arm, ribs, thoracic and lumbar spine, pelvis and legs). The same researcher completed analysis of defined body segments during the entire study period. Both the T- and Z-scores were calculated using gender and ethnic group-specific data from the national health and nutrition examination database (NHANES III).

### 2.3. Nutrition Intake and Analysis

Habitual dietary intake was assessed in 50 participants using a three-day food diary recorded over two weekdays and one weekend day [39]. At the point of handing out a blank food diary, the participants were also given in depth instructions on the level of detail to record daily food and drink intakes including meal time, food/ingredients weight and drinks volume, commercial brand names of food/ingredients and drink, any leftovers and cooking preparation methods. The participants were asked to maintain their normal eating habits over the three-day period. Dietary analysis was conducted using Nutritics software (version 1.8, Nutritics Ltd., Co. Dublin, Ireland) with one researcher completing all analyses. The participants’ total nutritional intake and the identified positive bone health related nutrients were scored against the recommended daily values [40,41] (see Table A2). An estimation of the participants’ metabolic balance (defined in Table A2) was ascertained using the Harris–Benedict equation [42], through the calculation of the participants’ basal metabolic rate when accounting for PA levels. This method of quantifying energy expenditure has been previously validated in mid-to-older aged adults [43].

### 2.4. PA Questionnaire

The participants’ PA status in a total of 50 participants (the same subsample who also completed the food diaries) was established using the Baecke PA questionnaire [44]. The questionnaire is split into three sections that denote work-, sport- and leisure-based PA and furthermore, give a combined score categorized as a global index of all these sub-sections. The participants who did not work due to retirement from their previous job were asked to fill in the work section as if their daily life/activities were their job. Each section was scored using a five-point scale and was calculated using a predetermined formula [44]. Work scoring focused on the physical intensity of working and factored in time spent sitting, whilst leisure scoring focused on leisure-based non-structured PA and factored in time spent watching television. Sport scoring denoted structured PA categorized by the intensity, repetition and duration of the activity undertaken.

### 2.5. Serum Inflammatory Cytokine Concentration

Prior to any physical testing, the same 50 participants who had provided food and PA data, were also asked, and consented, to the blood sampling. Our results include data from the 33 participants willing to provide the required 10 mL fasted (12 h) blood sample between 08:00 and 09:00, having not performed vigorous exercise for 48 h prior. Indeed blood samples were unobtainable for 17 participants due to either sampling failure or withheld consent. Blood was collected in anticoagulant-free vacutainers (BD Vacutainer Systems, Plymouth, UK) and rested on crushed ice for 10–15 min. The samples were then placed into a centrifuge (IEC CL31R, Thermo Scientific, Massachusetts, United States) for 10 min at 4000 rpm (2700× g), after which, serum was extracted and stored in 2 mL aliquots at −20 °C until subsequent analysis.

Multiplex luminometry was used to measure the serum concentrations of nine inflammatory cytokines (pro-inflammatory: interleukin (IL)-1β, IL-6, tumor necrosis factor (TNF)-α, Granulocyte-colony stimulating factor (G-CSF), interferon gamma (IFNg); anti-inflammatory: IL-10, transforming growth factor (TGF)-β1, β2 and β3) and five chemokines (IL-8, monocyte chemoattractant protein (MCP)-1, macrophage inflammatory protein (MIP)-1α, MIP-1β), regulated on activation, normal T cell expressed and secreted (RANTES). A 3-plex panel was used to measure TGF-β1, TGF-β2 and TGF-β3 concentrations (R&D Systems Europe Ltd., Abingdon, UK) and a Bio-Plex Pro Human Inflammation Panel Assay (Bio-Rad laboratories Ltd., Hemel Hempstead, UK) was used to measure the remaining 11 cytokines, following the manufacturer’s instructions. Samples were analyzed using a Bio-Plex 200 system (Bio-Rad laboratories Ltd., Hemel Hempstead, UK).

### 2.6. Statistical Analyses

The statistical analyses were carried out using SPSS (Version 22, SPSS Inc., Chicago, IL, USA). To determine parametricity (for adiposity, BMI, FMI, bone health), the Kolmogorov–Smirnov (whole sample *n* > 50) or the Shapiro–Wilk (if sub-sample *n* < 50) tests were utilized to determine if the sample was normally distributed. The Levene’s test was used to determine homogeneity of variance between groups. If parametric assumptions were met, between-group differences were examined by independent t-tests (for adiposity and bone health) or one-way ANOVA (for BMI and FMI) with post hoc pairwise comparisons conducted using the Bonferroni correction. However, if parametric assumptions were breached, between-group differences were examined by the Mann–Whitney U test (for adiposity) or a Kruskal–Wallis non-parametric ANOVA (for BMI and FMI) with post hoc pairwise comparisons being examined by Dunn correction. Pearson (or Spearman rank order for non-parametric data sets) bivariate correlations were used to define any associations between bone vs. age, PA scores, adiposity, BMI and nutritional variables, as well as serum cytokine concentration vs. bone health. Overall synthesis, including the radar graphs (Microsoft Excel, Version 2013 Washington, DC, USA), of participants habitual diet, participant characteristics and endocrine profile categorized by bone health (normal range vs. osteopenia) was computed through Z-scores (i.e., [mean of group – mean of sample population] ÷ standard deviation of sample population). Comparisons between Z-scores of the grouping variables were conducted by converting Z-scores into percentages using a Z-score comparison table. The calculation of unit-weighted Z-scores including the direction for habitual nutritional intake was done for all nutrients of interest. The unit-weighted Z-scores for participant characteristics including direction were calculated through positive signs for PA characteristics and lean mass, versus negative signs for age, BMI, body fat% and fat mass. Finally, unit-weighted Z-scores including direction were calculated for participants’ endocrine profiles using negative signs for IL-1β, IL-6, TNF-α, G-CSF, IFNg, IL-8, MCP-1, MIP-1α, MIP-1β and RANTES, versus positive signs for IL-10, TGF-β1, β2 and β3. Data are reported as mean (SD) and statistical significance was accepted when *p* ≤ 0.05.

## 3. Results

### 3.1. Descriptive Characteristics of Participants

Table 1 displays the descriptive characteristics of 163 untrained participants categorized by three different methods of classifying obesity: body fat%, BMI and FMI (Table 1). The descriptive characteristics, habitual nutritional intake, PA scores and endocrine profiles of the 50 untrained 43–80 years old middle-to-older aged sub-sample are reported in Table A1**,** where it was observed that there were no differences in the PA scores of the 50 participants between body fat%, BMI, FMI and bone health classifications.

### 3.2. Body Fat%, BMI and FMI Impact on Bone Mineral Content and Density

The positive effect of obesity on bone was demonstrated in all three classifications to differing degrees. BMI was found to have the greatest effect on bone properties through an increasing BMI classification being positively associated with 20/22 measured bone characteristics (Table 2). This was followed by an increasing FMI classification being positively associated with 14/22 bone characteristics and finally through a higher body fat% being positively associated with 9/22 positive bone characteristics (Table 2). The interpretation of these results would suggest that, as expected, BMI has the greatest loading effect on bone. Interestingly though, the effect of loading on bone appeared to be uniform across loaded (lumbar, pelvis and lower limbs) and unloaded (thoracic, ribs and upper limbs) bone sites (Table 2). The same pattern was continued to a lesser extent in the randomly selected 50 middle-to-older aged adults (as observed in Table A3). However, there was a reduction in the number of significant effects of the FMI on bone characteristics, which may be explained by a reduction in total mass due to lower lean mass in the older cohort.

When comparing the three definitions of obesity, classified by body fat%, BMI and FMI, utilizing Spearman rho correlations of the osteoporosis risk (T score) vs age, the only significant negative correlation observed was for obesity classified by body fat% (r = −0.43; *p* < 0.0061). These findings were confirmed in the middle-to-older age group, as a linear regression revealed only obese individuals classified by body fat% to be negatively associated with increasing age and T score (r = 0.46; r^2^ = 0.21; β = −0.084; *p* = 0.008).

Finally, of secondary note, the comparison of bone characteristics between the untrained and trained participants revealed the trained participants to have 7–50% significantly greater BMC and BMD characteristics at all body locations.

### 3.3. Habitual Dietary Intake

Analysis of participants’ habitual diet revealed that the entire sample consumes low amounts of trans fats (<2% daily total calories), with 98% of the participants also consuming below the recommended daily maximum intake for saturated fat (<11% daily total calories). Nutrients that are positively associated with bone health revealed 90% of participants achieved the recommended daily intake of calcium (>700 mg day), 84% met the requirements for zinc (male = ≥9.5 mg/day; female = ≥7 mg/day), 80% met the requirements for magnesium (male = ≥300 mg/day; female = ≥270 mg/day) and 100% met the requirements for phosphorus (≥550 mg). The incidences of the adequate intake of other bone-impacting nutrients of note that the participants achieved in their daily intake were vitamin C (94% participants), vitamin E (84% participants), vitamin K (14% participants), vitamin B-12 (100% participants), sodium (78% participants), omega-3 fatty acids (32% participants), omega-6 fatty acids (10% participants) and oligosaccharides (2% participants) (see Table A2 for both participants’ scoring and the criteria utilized). In other words, our sample’s diet was commendably good.

### 3.4. Bivariate Correlations

Table 3 displays the correlation coefficients between bone characteristics against the age, PA scores, indices of body composition and nutritional intake of 50 middle-to-older aged adults. Sport-based PA was revealed to be the most prolific predictor of bone structural characteristics with eight out of 12 significant positive associations, followed by BMI and total calorie intake with seven out of 12 significant positive associations. Age and body fat% revealed negative associations with 6/12 and 4/12 significant negative associations respectively, and global PA with three positive associations. Finally, adiposity revealed two significant positive associations and, surprisingly, bone nutrient score revealed two significant negative associations (Table 3). Surprisingly, independent analysis of macro and micronutrient intake between segmental BMD locations revealed significant associations including (a) Positive associations between vitamin A against the total BMD (*r* = 0.329; *p* = 0.020), thoracic BMD (*r* = 0.324; *p* = 0.022), lumbar BMD (*r* = 0.301; *p* = 0.034), pelvis BMD (*r* = 0.331; *p* = 0.019), dominant ribs (*r* = 0.329; *p* = 0.020) and the non-dominant ribs (*r* = 0.418; *p* = 0.002). (b) A negative association between relative protein intake vs. the dominant arm BMD (*r* = −0.330; *p* = 0.019) and non-dominant arm BMD (*r* = −0.359; *p* = 0.011). Aligned with our hypothesis, there was a significant positive association between relative protein intake vs. the non-dominant leg BMD (*r* = 0.418; *p* = 0.002). However, a partial correlation controlling for BMI removed this association between relative protein intake vs. the non-dominant leg BMD (*r* = −0.132; *p* = 0.364).

### 3.5. Serum Cytokine Concentrations vs. BMC and BMD

There were no significant associations between IFNg, IL-8, IL-10, TGFβ-1 and TGFβ-2 against a series of bone characteristics (BMC, BMD, T-score and Z-score), and/or 30 nutrition variables. However, the remaining nine cytokines and chemokines (G-CSF, TNFα, IL1β, IL-6, MCP-1, MCP-1β, MIP1α, RANTES, TGFβ-3) showed statistically significant associations; all were positive with the exception of RANTES and MCP-1 which were negatively associated (*p* < 0.05) or trend (*p* < 0.1) against BMC and/or BMD parameters (see Table 4).

### 3.6. Z-Score Comparisons of Nutrition Characteristics with Study Sample Classified by Bone Health

Figure 2A–C graphically summarize the overall dietary habits, characteristics and endocrine profiles of 50 participants grouped by their general bone health, utilizing their T-score to define osteopenia (T-score <−1.0) through the dimensionless variable Z-scores. Z-score differences between the relative and absolute protein intake between the bone health classifications differed, as illustrated by the similar absolute intake between groups, yet greater relative intake for those classified within the normal T-score range. This is demonstrated by the fact that the percentage difference between Z-scores for the absolute protein intake was only 0.4% between classification groups, thus interpreting this finding for the both groups to be well matched. However, when expressing protein intake relative to body mass (g/kg) the percentage difference between Z-scores was 10.2% with the normal range classified group having a lower relative protein intake. This pattern was continued for nutrients presumed beneficial to bone as demonstrated by calcium (−8.7%), phosphorus (−7.3%) and zinc (−10.5%), but differed for vitamin A (+23.3%), oligosaccharides (+20.2%), omega-6 fatty acid (+9.2%) intake between bone health classifications. Interestingly, individuals classified within the normal range had a 9.4% higher Z-score for the total calorie intake than those classified with osteopenia. The unit weighted Z-score for all nutrients classified from Figure 2A was calculated to be 0.118 (55%) for individuals who scored within a normal range T-score and –0.578 (28%) for individuals classified as osteopenic. Therefore, the difference in percentage between the unit-weighted Z-scores equated to 26.5% between bone health classifications.

In Figure 2B, the pattern of Z-scores of the participants’ characteristics revealed that the osteopenic participants have a higher percentage difference for age (+20.1%), BMI (+8.3%) and relative body fat content (+17.3%), but a lower lean mass (−22.5%), sport-based PA (−9.2%) and leisure-based PA (−9.5%). The unit-weighted score for participants’ characteristics, accounting for both positive and negative direction between osteopenic vs. normal range T-score adults, was 60.2% lower in the osteopenic group.

Finally, in Figure 2C, the pattern of Z-scores of the participants’ endocrine profiles revealed that the osteopenic participants have a higher percentage difference for IL-8 (+6.1%), RANTES (+13.4%) TGFβ-1 (+11.0%) and TGFβ-2 (+15.0%), but lower IFNg (−13.4%), IL-6 (−8.5%), IL-10 (−5.9%), IL-1β (−7.5%), TNFα (−15.1%), G-CSF (−23.5%), MIP-1α (−22.9%), MIP-1β (−11.6%) and TGFβ-3 (−17.7%). The unit-weighted score for the endocrine profiles, accounting for both positive and negative direction between osteopenic vs. normal range T-score adults, was 57.2% lower in the healthy bone group.

## 4. Discussion

The present study recognized key elements that influence BMD and potentially alleviate age-related BMC and BMD loss. These included varying combinations of optimizing total calorie intake, nutrient profile, sport-based PA body fat percentage, and BMI as we age. This was demonstrated by the osteopenic participants having a higher body fat%, undertaking less moderate to vigorous activity, whilst taking in lower total daily calories and participants with a healthy bone profile habitually consuming more oligosaccharides, omega-6 fatty acids and surprisingly also, vitamin A. The results thus support our first hypothesis and partially support our second. Interestingly, additional nutrients positively associated with bone health were not identified in individuals already within a healthy T-score range. However, our data should be contextualized in the fact that the greatest proportion of the study sample in fact habitually achieved the recommended intake for principal nutrients concerning bone health (calcium, zinc, magnesium and phosphorus). With regards to the third hypothesis, when participants were grouped by body fat% and FMI classification, the HA and obese individuals were not found to be negatively affected/disadvantaged by high adiposity with regards to either BMC or BMD and in fact demonstrated higher BMC and BMD in their non-dominant arm, thus rejecting our third hypothesis. Interestingly, individuals with a high BMI appeared to exhibit a loading response as demonstrated by significantly greater BMC and BMD in both their dominant and non-dominant lower limbs. However, in conjunction with this finding, it is interesting that this effect should also be seen to occur in their upper limbs. This latter observation would suggest that the healthier bone in high BMI adults in this age group is not just through additional mechanical loading. We propose that another, equally significant modulator of the greater bone health in high BMI individuals is the greater total calorie intake. Indeed, a covariate analysis correcting for dietary quantity removed the significant difference in bone health between BMI classifications. It is important to note that a strength of the current study design is that the sample was well matched with regards to PA for all group comparisons including between body fat%, BMI, FMI and bone health, and even gender grouping. Interestingly and in agreement with the current PA recommendations, we found that sport-based PA significantly positively correlated with the majority of bone sites (seven out of a possible ten and T score). The latter, as expected, was true on both dominant and non-dominant lower limb bone sites, thus supporting our fourth hypothesis. No correlations were observed between either work or leisure-based PA, which may be due to the age of the sample utilized and their current work status with the majority of individuals either retired or in part time work.

When analyzing the effect of nutrition on bone health in the current study, it was expected that particular nutrients already associated with good bone health would exhibit similar and positive correlations. Whilst, there were only two nutrients (namely, vitamin A and the relative protein intake) associated with BMD characteristics in bivariate correlation, others were highlighted as being important in the modulation of bone health through Z-score analyses including omega 6 fatty acids and oligosaccharides.

In the case of vitamin A, the current body of literature suggests that there is a U-shaped association with fracture risk [45,46]. Given that, within this study, we observed a series of positive correlations (*n* = 6) between vitamin A and a number of BMD sites, it would seem that our population, in terms of diet, was in the ascending limb of this U-shape relationship (<3000 µg [47]). The mechanism suggested for the positive association of vitamin A and fracture risk is via stimulation of osteoclast formation [48] and/or suppression of osteoblast activity [49], potentially through neutralizing the capability of VitD to maintain normal calcium levels [50]. On the other hand, the positive association between vitamin A and bone health is thought to be explained by vitamin A intake (Mean (SD): 1361 (1131 µg)) and not exceeding either upper limits of >3000 µg, where fracture risk increased by 48% when compared to individuals taking less than 1250µg [47], which was similar to our sample’s average intake.

Another initially surprising negative correlation was that between the relative protein intake and the BMD (dominant and non-dominant arm, and non-dominant leg). However, it is notable that, following a partial correlation controlling for BMI, this relationship was removed suggesting that the differences may have been attributed to the strong association between BMI and BMD. Protein intake is reported to positively influence not only musculoskeletal health (increasing or maintaining muscle mass) but it is also noted to play a role in bone [51]. The recommended intake for adults is the same for optimum musculoskeletal health starting at 0.8 g/kg body mass and rising to 1.2–1.6 g/kg body mass in elderly individuals [52]. The mean for the pooled study population was 1.17 g/kg body mass with 96% of the pooled sample achieving the recommended target intake of 0.8 g/kg body mass, demonstrating the high quality diet observed in this study population’s habitual lifestyle. This healthy dietary pattern is continued throughout the selected nutrients analyzed and may partially explain the lack of associations between nutritional variables and either the BMC or BMD characteristics. Interestingly, the best predictors of bone characteristics within the study was both diet quantity and quality, suggesting that adequate food consumption and quality is needed to ensure bone maintenance or growth can be achieved either through diet alone or in conjunction with structured PA. It should be noted that excess calorie intake above one’s metabolic demand may increase adipose tissue content and increase obesity risk.

The literature shows that obesity and bone health are negatively correlated, potentially through pro-inflammatory cytokines influencing the promotion of osteoclast activity [53] and bone resorption [7], thus negativity impacting bone characteristics. The pro-inflammatory cytokines IL-1β, IL-6, and TNF-α are important regulators of bone resorption and may play an important role in age-related bone loss [54]. Similarly, the TGF family plays a key role in bone homeostasis whereby therapies using these proteins seem to positively affect bone healing. Interestingly however, chronic inflammation (as normally expected in ageing and/or obesity) is associated with augmented levels of TGF-β1, and subsequently reduced bone mineral content and/or disturbed bone healing [55]. Overexpression of G-CSF (as seen in obesity for instance) induces severe osteopenia [56]. In parallel, IFNg stimulates osteoclast formation and hence bone loss via antigen-driven T-cell activation [57]. As for the anti-inflammatory cytokine IL-10, its deficiency is associated with osteopenia, decreased bone formation, and the mechanical fragility of bones [58]. On the other hand, high levels of IL-8 are associated with bone mineral accrual [59]. MCP-1 is thought to have beneficial effects on bone via stimulating the parathyroid hormone [60]. The MIP family has been associated with an acceleration of osteogenic differentiation and mineralization [61]. Last but not least, RANTES overexpression is associated with osteogenic differentiation [62]. Surprisingly, this was not observed within this study, as we noted positive associations (either correlation or trend) between four pro-inflammatory cytokines and three chemokines against both the BMD and BMC site locations (see Table 4). We would argue that our data demonstrate that, given the positive relationship between impact-based sport/exercise and bone health [63], the deleterious effects of concurrent high cytokines (TNFα, IL-1β, G-CSF, IL-6) and chemokines (MCP-1, MIP-1α, MIP-1β) was outweighed by the impact of a higher BMI adding much needed loading to the skeletal structure [64]. Whilst within our study adiposity appeared positively associated with both the dominant and non-dominant arm BMD, it is noteworthy that ~63% of the osteopenic participants were also high adipose. Therefore, whilst no negative association existed within this study, high levels of adiposity may instigate a poorer bone health, which may worsen with duration of exposure to obesity (number of years) and suboptimal diet, i.e., relatively low in bone health nutrients (see Figure 2A). However, in view of our findings and the limitations of our study, it is noted that blood samples were taken on a single occasion and were not taken over a course of a few months to confirm the average pro-inflammatory levels of each participant. Thus, future investigations should analyze the levels of vitamins and minerals within the blood alongside nutritional intake to examine the impact of nutrient deficiencies upon bone health and osteoporosis risk.

Finally, it is already widely accepted that PA is a preventative therapy for a number of deleterious ageing-related changes such as low skeletal muscle mass and strength [65], decreased physical function [66], and/or decreasing bone health [30,31]. Our data confirm these findings with regards to bone health, as noted by both classification of training status of participants and structured sport-based PA shown to correlate with 8/12 bone health variables. It is also noteworthy that the greatest impact of sport-based PA was in the loaded bone sites (both dominant and non-dominant legs and pelvis). With the benefits of PA reported to decrease the risk of hip fractures approximately by 20–40% [67,68,69] when compared to sedentary inactive individuals. Our findings lend further support to the association between increased PA and better BMD in vulnerable bone sites such as the hip and pelvis. Our study also demonstrates the importance of structured intense sport-based PA sessions in comparison to increasing either work or leisure-based PA as a tool to limit the risk of developing osteopenia or ultimately osteoporosis, with ageing. Ultimately, our data also suggest that extra calorie burning when performing sport-based PA in those with a higher BMI may be partly responsible for the increased bone mineral density and, counterintuitively, a relatively higher level of pro-inflammatory cytokine levels (due to the prolonged sport-based PA).

## 5. Conclusions

This study revealed total calorie intake, sport-based PA, BMI, adiposity, endocrine profile and age to be significant predictors of BMD characteristics in middle-to-older aged adults, with the main modifiable risk factor of developing osteoporosis being high body fat%. The analysis of nutritional profiles characterized by the participants’ bone health (normal vs osteopenia) revealed a pattern of positively associated nutrients related to bone health (omega-6 fatty acids, vitamin A and oligosaccharides) within the normal range group. Thus, application of this data suggests both the diet quantity and quality, supplemented with structured sport-based PA at a sufficient intensity for the intended age group, is associated with a healthy bone profile in later life. Future research should investigate how varying forms of PA impact on bone health to provide more prescriptive guidelines dependent on either age classification or existing bone health status.

## Figures and Tables

**Figure 1 nutrients-11-00195-f001:**
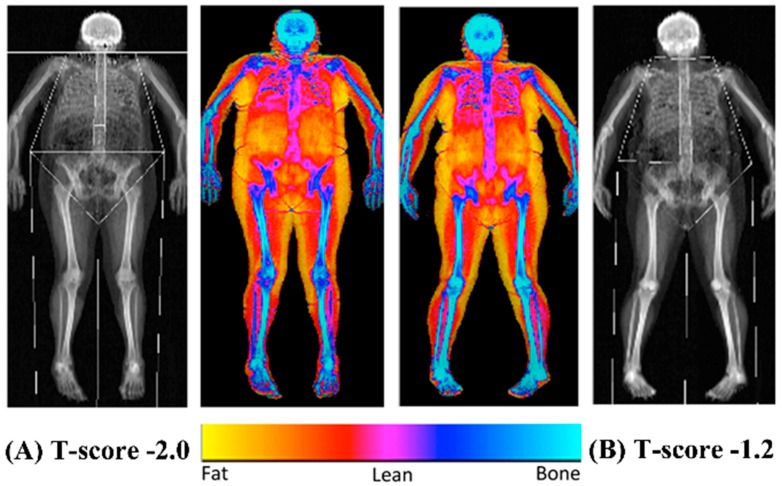
Representative dual energy x-ray absorptiometry scans of a female (**A**; T-score: −2.0) and male (**B**; T-score: −1.2) with osteopenia.

**Figure 2 nutrients-11-00195-f002:**
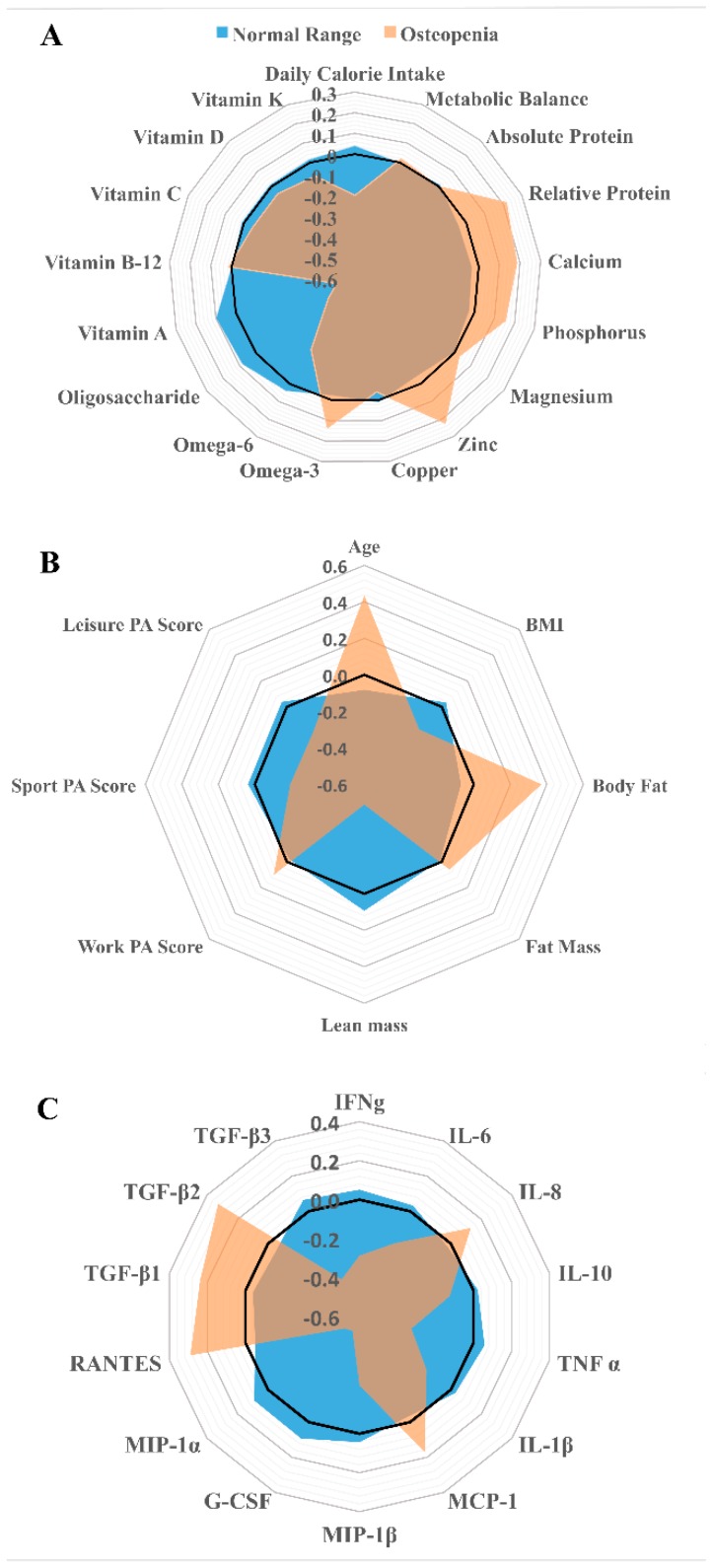
Comparison of the patterns of nutrient z-scores (**A**) associated with bone health [4] taken from the participants’ habitual diet, characteristics (**B**) and endocrine levels (**C**) and categorized by their T-score computed from reference data from the national health and nutrition examination database (normal range vs. osteopenia T- score <−1.0).

**Table 1 nutrients-11-00195-t001:** A total of 163 untrained participants’ anthropometric characteristics categorized by three methods of classifying obesity (body fat%, body mass index (BMI) and fat mass index (FMI)).

	Body Fat%		Body Mass Index				Fat Mass Index			
	NA (*n* = 83)	HA/Ob (*n* = 80)	U (*n* = 9)	NW (*n* = 53)	Ov (*n* = 54)	Ob (*n* = 47)	FD (*n* = 8)	NW (*n* = 53)	EF (*n* = 62)	Ob (*n* = 40)
**Characteristics**										
Age (years)	**38 ± 21 ^a^**	**49 ± 22 ^b^**	**34 ± 19 ^ab^**	**41 ± 22 ^a^**	**50 ± 23 ^b^**	**39 ± 19 ^ab^**	**36 ± 21 ^ab^**	**38 ± 21 ^a^**	**50 ± 23 ^b^**	**40 ± 19 ^ab^**
Height (m)	1.67 ± 0.09 ^a^	1.66 ± 0.08 ^a^	1.66 ± 0.06 ^a^	1.66 ± 0.09 ^a^	1.68 ± 0.09 ^a^	1.66 ± 0.08 ^a^	1.65 ± 0.07 ^a^	1.66 ± 0.09 ^a^	1.67 ± 0.09 ^a^	1.66 ± 0.08 ^a^
Body Mass (kg)	**65.5 ± 14.1 ^a^**	**85.4 ± 15.8 ^b^**	**50.0 ± 4.3 ^a^**	**60.3 ± 7.8 ^a^**	**77.1 ± 9.1 ^b^**	**94.8 ± 13.8 ^c^**	**50.3 ± 4.5 ^a^**	**61.3 ± 10.1 ^a^**	**77.0 ± 10.3 ^b^**	**96.1 ± 13.7 ^c^**
BMI (kg/m^2^)	**23.4 ± 3.8 ^a^**	**30.9 ± 3.8 ^b^**	**18.2 ± 0.6 ^a^**	**21.8 ± 1.6 ^a^**	**27.2 ± 1.2 ^b^**	**34.5 ± 4.5 ^c^**	**18.4 ± 1.0 ^a^**	**22.0 ± 2.4 ^a^**	**27.4 ± 2.0 ^b^**	**35.0 ± 4.7 ^c^**
Body Fat (%)	**30.3 ± 6.1 ^a^**	**41.5 ± 6.7 ^b^**	**25.8 ± 4.6 ^a^**	**30.4 ± 6.4 ^a^**	**36.2 ± 7.0 ^b^**	**43.3 ± 5.8 ^c^**	**23.7 ± 2.1 ^a^**	**29.7 ± 5.6 ^b^**	**37.1 ± 6.4 ^c^**	**44.3 ± 5.7 ^d^**
Fat Mass (kg)	**19.4 ± 6.0 ^a^**	**34.7 ± 9.6 ^b^**	**12.5 ± 2.2 ^a^**	**17.7 ± 3.8 ^b^**	**26.9 ± 4.1 ^c^**	**40.1 ± 8.6 ^d^**	**11.6 ± 1.3 ^a^**	**17.5 ± 3.4 ^a^**	**27.5 ± 4.2 ^b^**	**41.6 ± 8.4 ^c^**
FMI (kg/m^2^)	**7.0 ± 2.2 ^a^**	**12.7 ± 3.8 ^b^**	**4.6 ± 0.9 ^a^**	**6.5 ± 1.6 ^a^**	**9.6 ± 1.9 ^b^**	**14.7 ± 3.5 ^c^**	**4.3 ± 0.5 ^a^**	**6.4 ± 1.4 ^a^**	**9.9 ± 1.9 ^b^**	**15.3 ± 3.5 ^c^**
Lean Mass (kg)	**42.1 ± 11.0 ^a^**	**46.1 ± 9.4 ^b^**	**34.1 ± 4.1 ^a^**	**38.8 ± 7.3 ^a^**	**45.8 ± 9.7 ^b^**	**49.9 ± 8.6 ^c^**	**35.3 ± 3.7 ^a^**	**39.8 ± 8.5 ^a^**	**45.1 ± 9.7 ^b^**	**49.8 ± 8.6 ^b^**
Android Fat Mass (kg)	**1.43 ± 0.71 ^a^**	**3.14 ± 1.07 ^b^**	**0.72 ± 0.26 ^a^**	**1.23 ± 0.43 ^a^**	**2.29 ± 0.49 ^b^**	**3.74 ± 1.02 ^c^**	**0.58 ± 0.09 ^a^**	**1.20 ± 0.41 ^a^**	**2.36 ± 0.48 ^b^**	**3.92 ± 0.98 ^c^**
Gynoid Fat Mass (kg)	**3.61 ± 1.02 ^a^**	**5.88 ± 1.67 ^b^**	**2.61 ± 0.42 ^a^**	**3.33 ± 0.76 ^a^**	**4.73 ± 1.14 ^b^**	**6.69 ± 1.41 ^c^**	**2.47 ± 1.02 ^a^**	**3.31 ± 0.63 ^a^**	**4.82 ± 1.13 ^b^**	**6.90 ± 1.43 ^c^**
Android:Gynoid Ratio	**0.89 ± 0.19 ^a^**	**1.07 ± 0.15 ^b^**	**0.69 ± 0.12 ^a^**	**0.86 ± 0.19 ^a^**	**1.04 ± 0.17 ^bc^**	**1.08 ± 0.10 ^bc^**	**0.66 ± 0.08 ^a^**	**0.85 ± 0.17 ^a^**	**1.05 ± 0.17 ^b^**	**1.09 ± 0.10 ^b^**

Note: Data are the mean ± standard deviation. Group significant differences are highlighted in bold. Labelled Body Fat%, BMI and FMI pairwise means in a row without a common letter (^a b c d^) differ, *p* < 0.05. Non-parametric tests are highlighted in grey shading. Abbreviations: EF, Excess Fat; FD, Fat Deficit; HA, High Adipose; NA, Normal Adipose; NW, Normal Weight; Ob, Obese; Ov, Overweight; U, Underweight.

**Table 2 nutrients-11-00195-t002:** Bone mineral content (BMC) and bone mineral density (BMD) characteristics in 163 untrained participants categorized by body fat%, body mass index (BMI) and fat mass index (FMI) classifications.

	Body Fat%		Body Mass Index				Fat Mass Index			
	NA (*n* = 83)	HA/Ob (*n* = 80)	U (*n* = 9)	NW (*n* = 53)	Ov (*n* = 54)	Ob (*n* = 47)	FD (*n* = 8)	NW (*n* = 53)	EF (*n* = 62)	Ob (*n* = 40)
**BMC (g)**										
Total	2404 ± 499 ^a^	2532 ± 528 ^a^	**2000 ± 348 ^a^**	**2280 ± 401 ^a^**	**2603 ± 599 ^b^**	**2609 ± 443 ^b^**	**2123 ± 370 ^a^**	**2324 ± 457 ^a^**	**2540 ± 566 ^b^**	**2609 ± 469 ^b^**
Thoracic	**111 ± 28 ^a^**	**127 ± 32 ^b^**	**91 ± 16 ^a^**	**107 ± 22 ^a^**	**129 ± 37 ^b^**	**127 ± 28 ^b^**	**89 ± 18 ^a^**	**111 ± 29 ^b^**	**124 ± 32 ^c^**	**129 ± 30 ^c^**
Lumbar	66 ± 17 ^a^	66 ± 17 ^a^	59 ± 18 ^a^	64 ± 15 ^a^	69 ± 20 ^a^	67 ± 16 ^a^	62 ± 18 ^a^	65 ± 14 ^a^	68 ± 20 ^a^	67 ± 16 ^a^
Pelvis	267 ± 78 ^a^	260 ± 82 ^a^	**219 ± 69 ^a^**	**250 ± 68 ^a^**	**267 ± 93 ^a^**	**283 ± 74 ^a^**	228 ± 78 ^a^	259 ± 72 ^a^	262 ± 86 ^a^	280 ± 80 ^a^
**Dominant**										
Ribs	**95 ± 26 ^a^**	**115 ± 29 ^b^**	**74 ± 13 ^a^**	**87 ± 21 ^a^**	**108 ± 26 ^b^**	**128 ± 24 ^c^**	**79 ± 15 ^a^**	**88 ± 20 ^a^**	**106 ± 28 ^b^**	**130 ± 28 ^c^**
Arm	**162 ± 40 ^a^**	**178 ± 46 ^b^**	**132 ± 16 ^a^**	**156 ± 34 ^ab^**	**180 ± 50 ^bc^**	**181 ± 42 ^c^**	**140 ± 24 ^a^**	**159 ± 39 ^a^**	**175 ± 47 ^b^**	**182 ± 42 ^b^**
Leg	**452 ± 114 ^a^**	**479 ± 119 ^a^**	**363 ± 59 ^a^**	**424 ± 93 ^a^**	**494 ± 130 ^b^**	**499 ± 108 ^b^**	**391 ± 61 ^a^**	**435 ± 107 ^a^**	**481 ± 123 ^b^**	**495 ± 115 ^b^**
**Non-Dominant**										
Ribs	**96 ± 27 ^a^**	**108 ± 33 ^b^**	**72 ± 14 ^a^**	**87 ± 17 ^a^**	**107 ± 30 ^b^**	**118 ± 33 ^c^**	**77 ± 16 ^a^**	**88 ± 19 ^a^**	**106 ± 30 ^b^**	**117 ± 35 ^c^**
Arm	**154 ± 38 ^a^**	**170 ± 51 ^b^**	**123 ± 15 ^a^**	**147 ± 32 ^a^**	**175 ± 50 ^b^**	**172 ± 50 ^b^**	**131 ± 19 ^a^**	**151 ± 38 ^a^**	**169 ± 47 ^b^**	**173 ± 52 ^b^**
Leg	438 ± 121 ^a^	460 ± 129 ^a^	**352 ± 69 ^a^**	**407 ± 96 ^a^**	**486 ± 142 ^b^**	**471 ± 123 ^b^**	**378 ± 69 ^a^**	**418 ± 112 ^a^**	**471 ± 132 ^b^**	**469 ± 131 ^b^**
**BMD (g/cm²)**										
Total	1.190 ± 0.121 ^a^	1.206 ± 0.139 ^a^	**1.097 ± 0.076 ^a^**	**1.160 ± 0.108 ^a^**	**1.227 ± 0.156 ^b^**	**1.227 ± 0.111 ^b^**	1.136 ± 0.077 ^a^	1.173 ± 0.119 ^a^	1.209 ± 0.148 ^a^	1.225 ± 0.118 ^a^
Thoracic	**1.001 ± 0.147 ^a^**	**1.084 ± 0.175 ^b^**	**0.909 ± 0.083 ^a^**	**0.978 ± 0.123 ^ab^**	**1.060 ± 0.190 ^bc^**	**1.118 ± 0.153 ^c^**	**0.944 ± 0.075 ^a^**	**0.999 ± 0.135 ^a^**	**1.038 ± 0.188 ^a^**	**1.125 ± 0.151 ^b^**
Lumbar	1.001 ± 0.148 ^a^	1.149 ± 0.198 ^a^	1.068 ± 0.152 ^a^	1.125 ± 0.164 ^a^	1.161 ± 0.229 ^a^	1.177 ± 0.182 ^a^	1.118 ± 0.168 ^a^	1.136 ± 0.151 ^a^	1.144 ± 0.232 ^a^	1.178 ± 0.198 ^a^
Pelvis	1.001 ± 0.149 ^a^	1.266 ± 0.177 ^a^	**1.009 ± 0.095 ^a^**	**1.172 ± 0.156 ^a^**	**1.270 ± 0.187 ^b^**	**1.322 ± 0.161 ^b^**	**1.100 ± 0.132 ^a^**	**1.201 ± 0.163 ^a^**	**1.256 ± 0.184 ^ab^**	**1.310 ± 0.169 ^b^**
**Dominant**										
Ribs	0.711 ± 0.089 ^a^	0.713 ± 0.100 ^a^	**0.650 ± 0.062 ^a^**	**0.691 ± 0.081 ^a^**	**0.724 ± 0.116 ^a^**	**0.731 ± 0.078 ^a^**	0.690 ± 0.070 ^a^	0.702 ± 0.089 ^a^	0.711 ± 0.111 ^a^	0.729 ± 0.078 ^a^
Arm	**0.797 ± 0.132 ^a^**	**0.848 ± 0.154 ^b^**	**0.700 ± 0.025 ^a^**	**0.786 ± 0.127 ^ab^**	**0.842 ± 0.164 ^bc^**	**0.863 ± 0.133 ^c^**	**0.715 ± 0.052 ^a^**	**0.801 ± 0.155 ^ab^**	**0.829 ± 0.144 ^bc^**	**0.860 ± 0.132 ^c^**
Leg	1.209 ± 0.172 ^a^	1.254 ± 0.179 ^a^	**1.090 ± 0.052 ^a^**	**1.166 ± 0.180 ^a^**	**1.267 ± 0.177 ^b^**	**1.291 ± 0.148 ^b^**	**1.126 ± 0.061 ^ab^**	**1.183 ± 0.191 ^a^**	**1.245 ± 0.169 ^ab^**	**1.293 ± 0.159 ^b^**
**Non-Dominant**										
Ribs	0.716 ± 0.102 ^a^	0.714 ± 0.083 ^a^	**0.628 ± 0.064 ^a^**	**0.693 ± 0.085 ^ab^**	**0.730 ± 0.111 ^bc^**	**0.740 ± 0.067 ^c^**	0.672 ± 0.057 ^a^	0.702 ± 0.098 ^a^	0.718 ± 0.104 ^a^	0.736 ± 0.069 ^a^
Arm	**0.769 ± 0.119 ^a^**	**0.817 ± 0.128 ^b^**	**0.685 ± 0.035 ^a^**	**0.735 ± 0.075 ^a^**	**0.823 ± 0.149 ^b^**	**0.844 ± 0.115 ^b^**	**0.700 ± 0.036 ^a^**	**0.750 ± 0.103 ^a^**	**0.808 ± 0.141 ^b^**	**0.843 ± 0.112 ^c^**
Leg	1.200 ± 0.214 ^a^	1.251 ± 0.169 ^a^	**1.076 ± 0.061 ^a^**	**1.180 ± 0.145 ^ab^**	**1.268 ± 0.196 ^bc^**	**1.254 ± 0.233 ^c^**	**1.108 ± 0.077 ^a^**	**1.199 ± 0.159 ^ab^**	**1.225 ± 0.243 ^bc^**	**1.283 ± 0.149 ^c^**
**Z-score**	0.91 ± 1.11 ^a^	1.07 ± 0.98 ^a^	**0.03 ± 0.83 ^a^**	**0.67 ± 1.06 ^a^**	**1.25 ± 0.90 ^b^**	**1.25 ± 1.03 ^b^**	0.46 ± 0.55 ^a^	0.77 ± 1.11 ^a^	1.10 ± 0.97 ^a^	1.22 ± 1.08 ^a^
**T-score**	0.68 ± 1.22 ^a^	0.73 ± 1.32 ^a^	**−0.09 ± 0.91 ^a^**	**0.34 ± 1.17 ^a^**	**0.80 ± 1.32 ^ab^**	**1.17 ± 1.20 ^b^**	0.29 ± 0.77 ^a^	0.48 ± 1.20 ^a^	0.70 ± 1.34 ^a^	1.11 ± 1.27 ^a^

Note: Data are the mean ± standard deviation. Group significant differences are highlighted in bold. Labelled Body Fat%, BMI and FMI pairwise means in a row without a common letter (^a b c^) differ, *p* < 0.05. Non-parametric tests are highlighted in grey shading. Abbreviations: EF, Excess Fat; FD, Fat Deficit; HA, High Adipose; NA, Normal Adipose; NW, Normal Weight; Ob, Obese; Ov, Overweight; U, Underweight.

**Table 3 nutrients-11-00195-t003:** Bivariate correlations between habitual lifestyle factors against bone mineral density (BMD) characteristics in designated body locations in 50 43–80-year-old adults.

		Physical Activity Score								Nutrition		
BMD Location	Age	Work	Sport	Leisure	Global	Adiposity	Body Fat %	BMI	FMI	Daily Nutrition Score	Bone Score	Total Calorie Intake
Total	**−0.42 ****	0.10	**0.35 ***	−0.12	0.21	0.06	**−0.33 ***	0.26	−0.46	−0.16	−0.28	**0.40 ****
Thoracic	−0.20	0.10	0.23	−0.18	−0.03	0.25	0.06	**0.38 ****	0.21	−0.08	−0.07	0.22
Lumbar	−0.24	−0.12	**0.31 ***	−0.17	0.01	-0.06	−0.24	0.04	−0.10	−0.04	−0.09	0.15
Pelvis	**−0.45 ****	0.09	**0.40 ****	0.10	**0.36 ***	0.17	−0.07	**0.28 ***	0.10	−0.20	−0.27	0.19
**Dominant**												
Rib	−0.24	0.14	**0.37 ****	−0.03	0.20	−0.05	**−0.37 ****	0.13	−0.16	−0.04	−0.14	**0.52 *****
Arm	−0.19	0.11	0.26	−0.06	0.17	**0.31 ***	-0.11	**0.50 *****	0.18	−0.17	−0.23	**0.47 ****
Leg	**−0.28 ***	0.14	**0.35 ***	−0.09	0.25	0.18	−0.26	**0.39 ****	0.06	−0.16	−0.21	**0.48 *****
**Non-Dominant**												
Rib	**−0.31 ***	0.23	**0.34 ***	−0.05	**0.29 ***	0.07	**−0.31 ***	**0.32 ***	−0.02	−0.08	−0.19	**0.43 ****
Arm	−0.21	0.09	0.21	−0.05	0.16	**0.33 ***	−0.08	**0.53 *****	0.21	−0.13	−0.24	**0.44 ****
Leg	**−0.34 ***	0.24	**0.34 ***	−0.08	0.24	0.19	**−0.28 ***	**0.40 ****	0.06	−0.19	−0.25	**0.48 *****

**Z-score**	−0.11	0.02	0.24	0.07	0.18	−0.03	−0.18	0.06	−0.08	−0.24	**−0.28 ***	0.07
**T-score**	**−0.45 ****	0.05	**0.37 ****	0.01	**0.30 ***	0.08	−0.21	0.23	0.01	−0.19	**−0.33 ***	0.21

Note: Spearman rank order correlations highlighted in grey. Significant correlations are highlighted in bold (* *p* < 0.05, ** *p* < 0.01, *** *p* < 0.001). Abbreviations: BMI, Body Mass Index; FMI, Fat Mass Index.

**Table 4 nutrients-11-00195-t004:** Spearman rank order correlations between serum cytokine concentrations in 33 participants against participants’ bone characteristics.

	Correlation Coefficient (*r*)																					
	Bone Mineral Density												Bone Mineral Content									
					Dominant			Non-Dominant									Dominant			Non-Dominant		
	Total	Thoracic	Lumbar	Pelvis	Rib	Arm	Leg	Rib	Arm	Leg	T-score	Z-score	Total	Thoracic	Lumbar	Pelvis	Rib	Arm	Leg	Rib	Arm	Leg
**IL-1β**	0.25	0.16	0.10	0.13	0.05	0.15	0.25	0.22	0.16	0.29	0.26	**0.36 ***	0.27	0.04	0.16	0.22	−0.24	0.31	0.33	0.11	0.25	0.32
**IL-6**	0.22	0.20	−0.02	0.16	0.03	0.13	0.19	0.22	0.17	0.21	0.24	0.32	0.25	0.18	0.01	0.26	−0.15	0.27	0.24	0.12	0.26	0.24
**TNFα**	0.25	0.19	0.07	0.12	0.05	0.18	0.26	0.21	0.18	0.29	0.28	**0.35 ***	0.32	0.14	0.17	0.22	−0.07	0.33	**0.39 ***	0.09	0.27	**0.39 ***
**G-CSF**	0.27	0.05	0.04	0.19	0.26	0.07	0.12	0.21	0.10	0.20	0.31	0.29	0.22	0.26	0.17	0.23	0.13	0.11	0.26	0.18	0.08	0.26
**IFNg**	0.04	−0.07	−0.09	−0.16	0.07	0.12	0.04	0.02	0.06	0.02	−0.04	0.00	0.00	−0.12	0.05	−0.17	−0.05	−0.01	0.07	0.02	0.00	0.06
**IL-10**	0.13	0.16	0.00	0.11	0.02	0.09	0.13	0.20	0.11	0.13	0.16	0.26	0.20	0.15	−0.03	0.19	−0.21	0.22	0.17	0.06	0.18	0.14
**TGFβ-1**	0.02	−0.31	−0.06	0.05	−0.05	−0.10	−0.12	−0.22	−0.06	−0.07	−0.04	−0.06	−0.22	−0.05	0.04	0.09	−0.08	−0.11	−0.14	−0.16	−0.05	−0.05
**TGFβ-2**	0.05	−0.16	0.06	0.05	−0.16	0.02	−0.02	−0.20	0.03	0.00	−0.04	0.09	−0.17	−0.10	0.11	−0.01	0.02	0.01	−0.03	−0.19	0.09	0.04
**TGFβ-3**	**0.44 ***	**0.45 ***	0.25	**0.48 ****	0.235	**0.55 ****	**0.46 ***	0.27	**0.55 ****	**0.51 ****	0.36	0.27	0.334	**0.50 ****	0.27	**0.39 ***	**0.60 ****	**0.41 ***	**0.41 ***	**0.43 ***	**0.41 ***	**0.47 ***
**IL-8**	0.08	0.11	−0.12	−0.04	−0.08	−0.03	0.07	0.03	0.07	0.12	0.04	0.17	0.03	0.05	−0.17	0.07	−0.16	0.09	0.09	0.10	0.10	0.13
**MCP-1**	−0.13	−0.15	−0.26	−0.14	−0.29	0.00	−0.09	**−0.35 ***	0.09	−0.08	−0.24	−0.18	−0.30	−0.03	−0.19	−0.01	−0.11	−0.13	−0.16	−0.17	−0.07	−0.07
**MIP-1α**	**0.44 ***	0.20	0.16	0.30	**0.39 ***	0.22	0.32	**0.42 ***	0.29	**0.38 ***	**0.48 ****	**0.47 ****	**0.39 ***	0.24	0.21	**0.43 ***	0.09	**0.36 ***	**0.43 ***	0.34	0.31	**0.42 ***
**MIP-1β**	0.31	0.19	0.15	0.25	0.19	0.14	0.19	0.21	0.18	0.22	0.29	0.25	0.20	0.14	0.19	0.34	0.10	0.16	0.25	0.23	0.24	0.28
**RANTES**	−0.27	−0.30	−0.31	−0.22	−0.18	−0.18	−0.28	**−0.39 ***	−0.14	−0.26	−0.31	**−0.40 ***	**−0.35 ***	0.02	−0.21	−0.18	0.10	−0.31	−0.30	−0.12	−0.24	−0.22

Note: Significant correlations are highlighted in black box (* *p* < 0.05; ** *p* < 0.01) and trends (*p* < 0.1) are highlighted in grey box. Abbreviations: G-CSF= Granulocyte-colony stimulating factor; IFNg= interferon gamma; IL= interleukin; MIP= macrophage inflammatory protein; MCP-1= monocyte chemoattractant protein; TGF= transforming growth factor; TNF= tumor necrosis factor; RANTES= regulated on activation, normal T cell expressed and secreted.

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
