# Peer review of "Body Fat Percentage, Body Mass Index, Fat Mass Index and the Ageing Bone: Their Singular and Combined Roles Linked to Physical Activity and Diet"

_nutrients, 2019, doi:10.3390/nu11010195_

Round 1
Reviewer 1 Report
The authors performed a study of factors that influence bone mineral density (BMD) and content with aging. They found that total calorie intake, sport-based physical activity, BMI, adiposity, endocrine profile and age to be significant predictors of BMD. The study appears rigorous and the report is well-written and comprehensive.
Major Comment:
Increased bone health was not associated with a general decrease in pro-inflammatory cytokines. Prolonged exercise is associated with increased pro-inflammatory cytokines yet is still considered healthy. In the Discussion you state that subjects with increased BMI had higher bone mineral density and suggest this is due to both increased load and increased nutritional uptake. However, those with a higher BMI must use more calories performing activities of daily living (ADL) to move the greater mass of their bodies around. Could this extra calorie burning (pseudo-exercise) when performing ADL in those with a higher BMI be partly responsible for the increased bone mineral density and lack of a decrease in pro-inflammatory cytokine levels (due to the prolonged pseudo-exercise)? If so, is there a way to take this effect into account?
Minor Comments: Word choice
Line 29: identified of -> identified as
Line 279: can may -> may
Line 317: score with -> score
Line 434: intake -> intake and
Line 422: and, -> and
Major Comment:
R1.Q1. Increased bone health was not associated with a general decrease in pro-inflammatory cytokines. Prolonged exercise is associated with increased pro-inflammatory cytokines yet is still considered healthy. In the Discussion you state that subjects with increased BMI had higher bone mineral density and suggest this is due to both increased load and increased nutritional uptake. However, those with a higher BMI must use more calories performing activities of daily living (ADL) to move the greater mass of their bodies around. Could this extra calorie burning (pseudo-exercise) when performing ADL in those with a higher BMI be partly responsible for the increased bone mineral density and lack of a decrease in pro-inflammatory cytokine levels (due to the prolonged pseudo-exercise)? If so, is there a way to take this effect into account?
R1.Q1.R1_ We agree with our reviewer that this is indeed a potential explanation for the data finding. We added the following sentence on P4 L495-498: Ultimately also, our data suggest that extra calorie burning when performing sport based PA in those with a higher BMI may be partly responsible for the increased bone mineral density and counterintuitively, a relatively higher level of pro-inflammatory cytokine levels (due to the prolonged sport based PA).
R1.Q2. Minor Comments: Word choice
Line 29: identified of -> identified as
Line 279: can may -> may
Line 317: score with -> score
Line 434: intake -> intake and
Line 422: and, -> and
R1.Q2.R2_ All recommendations of word choices have now been amended and highlighted within the manuscript in yellow.

Reviewer 2 Report
The manuscript by Tomlinson et al. tried to identify the key mediators for bone health. The authors concluded that high body fat% would be the focus of osteoporosis risk with ageing.
In abstract, the authors described “14/22” or “20/22”. The authors should explain what “22” mean.
Because the authors analyzed several cytokines, it would be informative if the authors add brief introduction for each cytokines and its relationship between bone health.
For statics, the authors should explain why they used three different statistical analysis for parametricity.
In line 337, the authors described that “statistically significant associations (P<0.05) or trends (P<0.1) against BMC and/or BMD parameters (see Table 4)”. The authors should describe whether these were positive or negative associations.
In line 115 and 125, physical activity should be PA. The authors should revise similar mistakes throughout the manuscript.
The authors should describe the limitation of this study. The authors analyzed several cytokines only one time point. Furthermore, although the authors analyzed the nutritional intake, the authors did not analyze the blood levels of several vitamins or minerals. I think these points might be the limitation of this study.
Author Response
Reviewer 2
R2.Q1. In abstract, the authors described “14/22” or “20/22”. The authors should explain what “22” mean.
R2.Q1.R_ The abstract has now been amended to state that 22 means the number of BMC and BMD measured outcomes that were analysed
“Obesity classed via BMI positively affected 20 out of 22 BMC and BMD-related outcome measures, whereas FMI was associated with 14 outcome measures and adiposity only modulated 9 out of 22 BMC and BMD-related outcome measures.”
R2.Q2.Because the authors analysed several cytokines, it would be informative if the authors add brief introduction for each cytokines and its relationship between bone health.
R2.Q2.R_ As per our reviewer’s recommendation we have now added a paragraph summarising the link between each endocrine factor and bone health (see page 17 para 2 L454-467)
“The pro-inflammatory cytokines IL-1β, IL-6, and TNF-α are important regulators of bone resorption and may play an important role in age-related bone loss [54]. Similarly, the TGF family plays a key role in bone homeostasis whereby therapies using these proteins seem to positively affect bone healing. Interestingly however, chronic inflammation (as normally expected in ageing and/or obesity), is associated with augmented levels of TGF-β1, and subsequently reduced bone mineral content and/or disturbed bone healing [55]. Overexpression of G-CSF (as seen in obesity for instance) induces severe osteopenia [56]. In parallel IFNg stimulates osteoclast formation and hence bone loss via antigen driven T-cell activation [57]. As for the anti-inflammatory cytokine IL-10, it deficiency is associated with osteopenia, decreased bone formation, and mechanical fragility of bones [58]. On the other hand, high levels of IL-8 are associated with bone mineral accrual [59]. MCP-1 is thought to have beneficial effects on bone via stimulating the parathyroid hormone [60]. The MIP family has been associated with an acceleration of osteogenic differentiation and mineralization [61]. Last but not least, RANTES overexpression is associated with osteogenic differentiation [62].”
R2.Q3. For statistics, the authors should explain why they used three different statistical analyses for parametricity.
R2.Q3.R_ This has now been clarified within the statistical analyses section of the manuscript as to why each method was used and what they were testing (see L209-210).
“Statistical analyses were carried out using SPSS (Version 22, SPSS Inc., Chicago, IL, USA). To determine parametricity (for adiposity, BMI, FMI, bone health), Kolmogorov–Smirnov (whole sample n>50) or Shapiro–Wilk (if sub-sample n<50) were utilized to determine if the sample was normally distributed and Levene’s tests to determine homogeneity of variance between groups.”
R2.Q4.In line 337, the authors described that “statistically significant associations (P<0.05) or trends (P<0.1) against BMC and/or BMD parameters (see Table 4)”. The authors should describe whether these were positive or negative associations.
R2.Q4.R. We appreciate our reviewer comment and we now specify the direction of the associations (see L338-339).
“There were no significant associations between IFNg, IL-8, IL-10, TGFβ-1 and TGFβ-2 against a series of bone characteristics (BMC, BMD, T-score and Z-score), and/or 30 nutrition variables. However, the remaining 9 cytokines and chemokines (G-CSF, TNFα, IL1β, IL-6, MCP-1, MCP-1β, MIP1α, RANTES, TGFβ-3) showed statistically significant associations, all positive with the exception of RANTES and MCP-1 which were negatively associated (P<0.05) or trends (P<0.1) against BMC and/or BMD parameters (see Table 4).”
R2.Q5. In line 115 and 125, physical activity should be PA. The authors should revise similar mistakes throughout the manuscript.
R2.Q5.R. The authors apologise for this oversight but all reference to physical activity post being defined as PA has now been amended within the manuscript and highlighted in yellow.
R2.Q6. The authors should describe the limitation of this study. The authors analyzed several cytokines only one time point. Furthermore, although the authors analyzed the nutritional intake, the authors did not analyze the blood levels of several vitamins or minerals. I think these points might be the limitation of this study.
R2.Q6.R. Thank you for your comment and the study limitation has now been added to discussion section and highlighted (see Lines 478-483).
“However, in view of our findings and limitation of our study, it is noted that blood samples were taken on a single occasion and were not taken over a course of a few months to confirm the average pro-inflammatory levels of each participant. Thus, future investigations should analyze the levels of vitamin and minerals within the blood alongside nutritional intake to examine the interactive of any potential nutrient deficiencies have upon bone health and osteoporosis risk.”

Round 2
Reviewer 2 Report
The authors fully answered my questions.